# Proximity can induce diverse friendships: A large randomized classroom experiment

**Julia M. Rohrer** [1,2]*, **Tamás Keller**[3,4,5], **Felix Elwert**[6]

**1** Department of Psychology, University of Leipzig, Leipzig, Germany, **2** International Max Planck Research School on the Life Course (LIFE), Max Planck Institute for Human Development, Berlin, Germany, **3** Computational Social Science - Research Center for Educational and Network Studies, Center for Social Sciences, Budapest, Hungary, **4** Institute of Economics, Center for Economic and Regional Studies, Budapest, Hungary, **5** TÁRKI Social Research Institute, Budapest, Hungary, **6** Department of Sociology & Department of Biostatistics and Medical Informatics, University of Wisconsin-Madison, Madison, Wisconsin, United States of America

* julia.rohrer@uni-leipzig.de

**Data Availability Statement:** The data are available on the Open Science Framework (https://osf.io/4vjc5/).

**Funding:** This research is funded by a grant from the Hungarian National Research, Development

## Abstract

Can outside interventions foster socio-culturally diverse friendships? We executed a large field experiment that randomized the seating charts of 182 3rd through 8th grade classrooms ($N$ = 2,966 students) for the duration of one semester. We found that being seated next to each other increased the probability of a mutual friendship from 15% to 22% on average. Furthermore, induced proximity increased the latent propensity toward friendship equally for all students, regardless of students' dyadic similarity with respect to educational achievement, gender, and ethnicity. However, the probability of a manifest friendship increased more among similar than among dissimilar students—a pattern mainly driven by gender. Our findings demonstrate that a scalable light-touch intervention can affect face-to-face networks and foster diverse friendships in groups that already know each other, but they also highlight that transgressing boundaries, especially those defined by gender, remains an uphill battle.

## Introduction

Friendships matter because social networks shape outcomes ranging from health behaviours to criminal activity and socio-economic achievement [1–7]. Friendship networks, however, are strongly constrained by homophily: Humans have the well-documented tendency to form and maintain relationships with those who resemble themselves along dimensions such as gender, race, ethnicity, age, religion, and education (e.g., [8]). Thus, social networks often lack diversity, and their benefits are distributed unevenly.

Previous research suggests that proximity between individuals can lead to friendships. In the present study, we investigate to which extent proximity can lead to friendships that transgress group boundaries imposed by homophily. For this purpose, we conducted a large-scale field experiment that randomly seated students in 3rd to 8th grade classrooms next to each other (Hungarian primary education, equivalent to ISCED 1 and ISCED 2 levels according to

and Innovation Office (NKFIH), Grant number: FK 125358 to Tamás Keller, and by a Vilas Faculty Mid-Career Award from the University of Wisconsin-Madison to Felix Elwert. The support from the János Bolyai Research Scholarship of the Hungarian Academy of Sciences and from the New National Excellence Program (ÚNKP) of the Ministry of Human Capacities are acknowledged (Grant number: ÚNKP-19-4-BCE-07).

**Competing interests:** The authors have declared that no competing interests exist.

the international classification). To our knowledge, this is the first randomized experiment of this kind outside of the elite context of college, and the first study to explicitly investigate whether the effects of proximity on friendship are modified by similarity along the lines of gender, educational achievement, and ethnicity.

## What do we know about the effects of proximity?

Prior research on the causal effects of proximity on friendships has focused on college freshmen. Multiple studies exploited natural experiments, such as alphabetical seating [9, 10]. Others directly randomized the assignment of college roommates [11], or the seating chart during an introductory meeting of psychology freshmen [12]. Most of these studies find that proximity promotes friendship formation. Some studies further assessed downstream outcomes of induced proximity, establishing positive effects of being randomly assigned a Black roommate on White students' attitudes and behavior [11, 13–18].

## Open questions

Prior findings are promising but leave two important questions open. First, prior work studied college freshmen, a highly selected and comparatively homogeneous elite in unusual circumstances, especially if they live in dorms. Having recently relocated from the parental home to college, college students must quickly construct a new social network among strangers from scratch [19] and thus may be particularly susceptible to the effects of proximity. This raises the question whether previous findings generalize to other populations in more quotidian and scalable settings. The only recent (non-randomized) study on the effects of being seated next to each other on friendships among school-age children failed to find an effect of proximity on friendship nominations [20, 21].

Second, little is known about the boundary conditions of the effects of proximity. Previous studies have addressed whether or not proximity can lead to inter-ethnic friendships (in the college context), but have neglected other potentially relevant dimensions of socio-cultural diversity. For example, it is unknown whether proximity can lead to friendships among students with different levels of educational achievement, which may expose lower-achieving students to positive role models; or whether proximity can lead to mixed-gender friendships, which may discourage the development of gendered attitudes and communication styles that are linked to power asymmetries in adulthood [22–24].

## The present study

To investigate the effects of proximity on friendship in general, and the extent to which proximity can promote boundary-crossing friendships among school-age children in particular, we conducted a large pre-registered field experiment. We randomized the seating charts within 182 3rd through 8th grade classrooms in 40 primary schools in rural Hungary for the duration of the Fall 2017 semester (5 months; ordinarily, the majority of seating charts would be designed by teachers, see S2 Text). We assessed best-friend nominations at the beginning of the subsequent Spring 2018 semester to test the expectation that being seated next to each other had a positive causal effect on friendships (deskmate hypothesis). In contrast to previous studies among college students, we thus investigated the effects of proximity in a scalable environment (because nearly all children must attend school), at younger ages, and in groups that already know each other well (from 1st grade onwards).

Since humans tend to form friendships with self-similar others, the friendship-inducing effect of proximity is likely tampered by homophily. We therefore expect that induced proximity should promote friendship especially among individuals who resemble each other

(modification-by-similarity hypothesis). In this study, we investigate effect modification by students' dyadic similarity along the three salient dimensions of gender, educational achievement, and ethnicity (Roma/non-Roma) to quantify the extent to which proximity can promote diverse friendships with respect to these categories.

## Materials and methods

### Pre-registration

We follow a detailed pre-analysis plan, filed at the RCT registry of the American Economic Associations before the receipt of any outcomes data on April 13, 2018 (see https://doi.org/10.1257/rct.2895-1.0). Deviations from the plan are explained in S1 Text and on the project page on the Open Science framework (https://osf.io/4vjc5/), which archives the data necessary to reproduce our central analyses, all analytic scripts and more detailed results.

### Study overview

We executed a large-scale field experiment in primary schools in Hungary (általános iskola). Classroom-seating charts were randomized for the duration of the fall semester (September 2017 through January 2018). Outcome variables were collected through student surveys between February and April 2018. The analytic sample for the central analyses consisted of $N = 2,996$ students (forming 24,962 dyads) within 182 3rd through 8th grade classrooms at 40 schools. Of these students, 48.2% ($N = 1,447$) were female; and 22.2% were of Roma ethnicity ($N = 666$). Ethnicity was missing for 4.5% of the sample ($N = 136$). Students' ages ranged from 8 to 17 years ($M = 11.88$, $SD = 1.80$); the high maximum age is due to students who had to repeat classes. Class sizes ranged from 10 to 33 students ($M = 19.42$).

### Recruitment and sample

In the spring of 2017, we contacted all primary schools in 7 contiguous counties of central Hungary, excluding the capital city of Budapest, via the heads of the local school districts to elicit information about classroom layouts and seating practices. We aimed to enroll all 3rd through 8th grade classrooms in which (1) teachers would implement our randomized seating chart in three core subjects: Hungarian literature, Hungarian grammar, and mathematics; (2) all students would receive instruction in these subjects together (e.g., no ability grouping); (3) the classroom layout would form a grid of freestanding forward-facing 2-person desks.

   After obtaining initial participation agreements from principals and teachers at 55 schools and dropping schools and classrooms that did not meet our inclusion criteria (see pre-analysis plan for details), the pre-analysis plan anticipated a sample of $N = 3,814$ students across 195 classrooms at 41 schools. The pre-analysis plan also stipulated additional exclusion criteria going forward. Following these pre-registered criteria, we dropped (in this order) 13 classrooms (containing 226 students) in which fewer than 30% of students answered the friendship-nomination item; 391 students who did not answer the friendship-nomination item; 36 students with missing values on at least four of the seven variables comprising the similarity index; and 113 students who were assigned to sit alone (as a robustness check, we also report models including those students). Subsequent data inspection resulted in the exclusion of 11 doubly entered students, 5 students whose classrooms were smaller than the pre-registered minimum size of 10, and 36 students who had left their classrooms before the intervention. The final analytic sample consisted of $N = 2,996$ students forming 24,962 dyads within 182 3rd through 8th grade classrooms at 40 schools.

Pre-registered balance checks reported in S3 Text indicate excellent balance on all baseline covariates in the final analytic sample.

## Intervention and exposure variable

Before the start of the fall semester 2017, we randomly assigned the students within each classroom to freestanding forward-facing two-person desks via unconstrained random partitioning, using a random number generator. We based the randomization on the class rosters from the preceding spring semester and stipulated a replacement algorithm to account for the small number of students who would exit or enter the class roster during the summer, with the aim of plausibly preserving randomization. Teachers were instructed to fill the seats of exited students with entering students from left to right, front to back, in alphabetic order of entering students' surnames. We call the seating chart resulting from randomization and algorithm-compliant replacements the "intended seating chart." The intended seating chart underlies all of our analyses (intention-to-treat analyses). Our central experimental exposure variable is coded = 1 for each dyad within a classroom that comprises deskmates in the intended seating chart, and = 0 otherwise.

Teachers were instructed to employ the intended seating chart in the three core subjects of the curriculum—mathematics, Hungarian literature, and Hungarian grammar—from the first day of classes (September 1, 2017) until the end of the fall semester (January 31, 2018). These three subjects form the core of the curriculum and receive the greatest weight in admission to selective secondary schools. They were taught in the same room for all grade levels and accounted for 6 to 10 lessons per week (25 to 45 percent of all lessons). Enforcing the seating chart across all subjects was not possible because (1) in some subjects, classrooms were split into smaller groups (e.g., different foreign languages), and (2) in some subjects, students were not seated in a fixed grid-layout (e.g., physical education and arts). However, seating charts typically apply to all subjects in a given room, and (depending on the grade level) most subjects were taught in the same room. Thus, students assigned to sit next to each other in the three core subjects likely also sat next to each other in other subjects; but we did not verify adherence outside of the core subjects.

Teachers were permitted to reseat students, but were asked to preserve deskmate assignments wherever possible. For example, if a student had to be moved to the front of the classroom because of vision problems, we asked that her deskmate be moved with her. We assessed compliance via teacher reports after the second week of classes and via classroom visits by our field-staff throughout the fall semester; 94.4 percent of the dyads in which students actually sat next to each after the second week of classes comprised students who were supposed to sit next to each in the intended seating chart.

## Baseline covariates and similarity index

At the beginning of the study, classroom teachers reported students' characteristics including student's gender (male vs. female); ethnicity (Roma Hungarian vs. non-Roma Hungarian); and end-of-semester grades for spring 2017 in Hungarian literature, Hungarian grammar, mathematics, diligence and behavior. Grades ranged from 1 (worst) to 5 (best) for literature, grammar, and mathematics; and from 2 to 5 for diligence and behavior. From these fives grades, we generated the grade point average (GPA). Following the pre-analysis plan, missing teacher reports of baseline covariates (3.2 to 3.6% of the total sample) were filled in with students' self-reports collected at endline.

To quantify the similarity between students, we calculated Gower's general coefficient of similarity [25] for each dyad within a classroom. Gower's index is a simple metric to quantify

the similarity between two units along a number of variables that may be qualitative and/or quantitative. We calculated the similarity index based on students' gender, ethnicity, and baseline grades in literature, grammar, mathematics, behavior, and diligence. Gender and ethnicity were each weighted by a factor of 1; every baseline grade was weighted by a factor of 1/5 (i.e., all grades together received a weight of 1). Gower's index ranges from 0 (maximum possible dissimilarity along all dimensions) to 1 (perfect similarity along all dimensions). For example, a Roma girl and a non-Roma boy whose grades are all at opposing ends of the scale would receive a similarity index of 0; two Roma girls with exactly the same grades would receive a similarity index of 1. We standardized the similarity index for further analysis.

## Friendship nominations

At the end of the study, students were asked to nominate up to 5 of their "best friends" within the classroom as part of a written 45-minute in-class survey. In this study, the primary outcome is students' reciprocated friendships, coded = 1 if both dyad members nominated each other as best friends and = 0 otherwise. As robustness checks, we also analyzed best-friend nominations within the classroom regardless of reciprocation.

## Statistical analyses

**Deskmate hypothesis.** We begin by evaluating the hypothesis that induced proximity fosters friendship between students. The deskmate effect is the net effect of sitting next to each other on friendship formation and friendship dissolution: students who were seated next to each other may either be induced newly to form a reciprocated tie, or to forego the dissolution of an existing tie.

We model the effect of sitting next to each other on reciprocated friendship nominations using a Bayesian multi-membership multilevel probit model. This is a dyad-level model with one observation for each unordered dyad consisting of students $i$ and $j$ in classroom $c$,

$$Friendship^*_{\{ij\}c} = b_0 + b_D * Deskmate_{\{ij\}c} + Classroom_c + \sum_{s \in \{i,j\}} Student_{sc} + \epsilon_{\{ij\}c} \qquad (1)$$

where $Friendship^*_{\{ij\}c}$ is the latent continuous friendship propensity of the dyad; $Deskmate_{\{ij\}c} = 1$ if the students in the dyad are deskmates and = 0 otherwise; $Classroom_c$ is a vector of classroom fixed effects to account for randomization within classrooms; and $\epsilon_{\{ij\}c} \sim N(0, \ \sigma^2_\epsilon)$ is the i.i.d. dyad-specific error term. The term $\Sigma_{s \in \{i,j\}} Student_{sc}$ refers to the two i.i.d. random effects for the students $s$ in the dyad, $Student_{sc} \sim N(0, \sigma^2_{Student})$. The latent continuous friendship propensity is linked to manifest friendship via the threshold function $Friendship_{\{ij\}c} = 1$ if $Friendship^*_{\{ij\}c} > 0$ and = 0 otherwise.

We interpret the results of this model in two complementary ways to assess the causal effect of sharing a desk: (i) on the latent continuous friendship propensity, $Friendship^*_{\{ij\}c}$, and (ii) on the probability of a manifest binary friendship nomination, $Friendship_{\{ij\}c}$. First, following preferred practice in certain research fields (e.g., psychology), we present $b_D$, the probit coefficient for the deskmate indicator, which estimates the effect of sharing vs. not sharing a desk on the latent propensity for forming a reciprocated friendship. As always in probit (or logit) models, this parameter is identified only up to scale [26], so that the coefficient should only be interpreted qualitatively as evidence about the direction of the deskmate effect on the latent friendship propensity.

Second, following preferred practice in other fields (e.g., economics and sociology), we present the average marginal effect (AME) of sharing vs. not sharing a desk on the probability of forming a manifest reciprocated friendship. Since probit models are non-linear probability

models, the effect of deskmate exposures on the probability of manifest friendship nominations likely varies across dyads. The AME is the average of the effects of sitting vs. not sitting next to each other on the probability of forming a reciprocated friendship across all dyads. For clarity, we present AMEs alongside the average of the predicted probabilities (predictive margins) for forming a reciprocated friendship if all dyads were deskmates and if all dyads were not deskmates.

We evaluate the evidence for the deskmate hypothesis by computing Bayesian 95% credible intervals ($CI_{95}$) around our point estimates (probit coefficients, AMEs, and predictive margins, respectively). Bayesian $CI_{95}$ are defined as the intervals into which the unobserved parameter falls with 95% probability, incorporating information from the Bayesian priors. While there are fundamental differences between frequentist and Bayesian statistics, from a pragmatic perspective, Bayesian credible intervals and frequentist confidence intervals often lead to very similar numerical results [27]. We interpret credible intervals that exclude zero as evidence for the presence of an effect.

Importantly, since seating charts were randomized within classrooms, the estimated effect of sharing a desk—evaluated either as the effect on the latent friendship propensity or as the AME on the probability of manifest friendships—has a causal interpretation after controlling for classroom fixed effects.

**Modification-by-similarity hypothesis.** Next, we evaluate the hypothesis that the causal effect of sharing a desk on friendship formation increases when students resemble each other more with respect to baseline characteristics. Since dyadic similarity was not randomized, our analyses do not identify the causal effect of similarity on friendship. Instead, we estimate (i) observational homophily, i.e., the extent to which similar individuals happen to befriend each other regardless of sharing a desk, and (ii) effect modification, i.e., the extent to which the causal deskmate effect varies with observed dyadic similarity (see [28] on the difference between causal interaction and effect modification). The model evaluating observational homophily adds Gower's (1971) index of similarity to the model of Eq (1). The model evaluating the similarity hypothesis (i.e., effect modification) further adds the product term of the index of similarity and the deskmate indicator.

For graphical representations, we define "low" ("high") values of Gower's similarity index as values falling 1 SD below (above) the sample mean across all dyads.

To explore which specific dimensions of the similarity index drive heterogeneity in the deskmate effect, we also estimate secondary models that allow the deskmate effect to vary with indicators of the dyad's gender constellation (boy-boy, girl-girl, mixed-gender), ethnic constellation (both Roma, both non-Roma, mixed), or GPA. Analyses of modification by GPA incorporate two variables, the absolute GPA difference within the dyad and the dyad's mean GPA (to control for grade levels in order to isolate the role of grade differences). The analyses follow the same logic as for the index of similarity (i.e., we first add the indicator of similarity on the respective dimension to evaluate observational homophily, and then we further add its product term with the deskmate indicator to evaluate the modification-by-similarity hypothesis).

Special care must be taken when interpreting the coefficients of the product terms between the deskmate indicator and similarity measures for evidence about the modification-by-similarity hypothesis. Qualitative conclusions about interaction effects in non-linear models, such as probit or logit models, can depend on the scale of the outcome [29, 30]. For example, two variables that relate to the outcome additively on one scale may relate to the outcome multiplicatively after the model undergoes a non-linear transformation that changes the scale of the outcome. Hence, when evaluating effect modification or interactions between variables in a probit model, it is possible that two variables (here, the deskmate indicator and dyadic similarity) statistically interact in their effect on the latent continuous outcome (here, the latent

friendship propensity) but do not interact in their effect on the probability of the manifest binary outcome (here, the probability of friendship nominations), or vice versa—and this can result simply from mechanically transforming probit coefficients into AMEs *after* a given model has been estimated on given data.

Despite strong, and at times conflicting, preferences across methodological communities [29, 31–33], no outcomes scale is inherently superior to another. Analysts who are interested in how the effect of sharing a desk on the latent friendship propensity is modified by similarity would inspect the probit coefficient on the interaction between the deskmate indicator and the similarity measure. This would make sense, for example, for analysts who want to know whether all groups of students are similarly nudged toward more (or less) positive relations. By contrast, analysts who are interested in effect modification in the effect of sharing a desk on manifest friendship nominations would compare group-specific AMEs. This would make sense, for example, if analysts believe that sharp classifications into "friend" vs. "not a friend" matter for classroom dynamics. Since we are interested in both qualities of the friendship network (latent friendship propensities and manifest nominations), we present probit coefficients and AMEs (accompanied by the relevant predictive margins) alongside each other.

As before, we statistically evaluate our estimates (probit coefficients, AMEs, and predictive margins, respectively) by computing the relevant Bayesian $CI_{95}$. Additionally, in order to evaluate whether there is evidence for any variation between multiple groups (e.g., those defined by Gower's index, or by gender, ethnicity, or GPA constellations), we compare models with and without deskmate-by-group interactions by inspecting the difference in their expected predictive accuracies, $Diff_{ELPD}$ (a Bayesian measure of model fit), computed via approximate leave-one-out cross-validation. Following convention, we conclude in favor of a model if $Diff_{ELPD}$ is at least twice its standard errors, $Diff_{ELPD} \geq 2 * SE_{DiffELPD}$).

**Robustness checks.** We explored several robustness checks for the deskmate and similarity hypotheses. First, in addition to reciprocated friendships, we also analyzed friendship nominations regardless of reciprocation. These analyses only differed from the previously described models in that (i) they included twice as many dyads, because every unordered dyad corresponds to two ordered sender-receiver dyads; and (ii) they included random effects for senders and receivers. In contrast to the analysis of reciprocated friendships, lower-level units (dyads) where thus nested within one higher-level unit of the classification *sender* and one higher-level unit of the classification *receiver*, resulting in a cross-classified multilevel probit effect model.

Second, we address two methodological concerns (especially in economics) about probit fixed effects models, such as the models introduced above. First, it is known that non-linear fixed effects models can be problematic in small panels (here, small classrooms) [34]. To address this concern, we re-estimated our primary models by substituting class-size indicators for the classroom fixed effect, thus replacing our pre-registered (and potentially problematic) fixed-effects model with a more conventional covariate-adjusted model. This substitution is permissible since the fixed effect is only needed to control for differences across classrooms in the probability that a given dyad is a deskmate dyad. Since this probability only depends on class size, controlling for class size is sufficient for causal identification. Second, to address the more general skepticism about non-linear models in parts of the social sciences, we estimated linear probability models (LPMs) for the probability of manifest friendship nominations. Among other advantages, LPMs, in contrast to probit and logit models, do not rest on distributional assumptions about the structural errors of the latent continuous friendship propensity. These LPMs mirror the specification of the main analyses described above.

In short, results for all three types of models (probit with fixed effects, probit controlling for classroom size, LPM with fixed effects) were extremely similar. The largest absolute difference

in the estimated AMEs across models was 1.6 percentage points, with most discrepancies well below 1 percentage points, which does not affect our qualitative conclusions.

We present additional explanations behind all analyses, all model outputs, and a table contrasting the resulting estimates across different model specifications on the Open Science Framework.

**Attrition.** About 10% of students were omitted from our main analysis, because they did not provide friendship nominations (e.g., because they lacked parental consent for the endline survey, did not attend school on the day of the assessment, or skipped the question). Multivariate non-response models indicated some selective non-response. While gender and ethnicity did not predict missingness ($p >.12$); a 1 SD increase in GPA predicted a 2.4 percentage point increase in the probability of response ($p = .001$); and a 1 SD increase in similarity (Gower's index) predicted a small but statistically significant decrease of 1.5 percentage points in the probability of response ($p = .004$). To address possible bias from selective attrition, we ran two additional sets of analyses.

First, we estimated a lower bound for the deskmate effect by imputing missing friendships nominations under extremely conservative assumptions: whenever nominations were missing, we assumed that (1) the student did not nominate their deskmate and (2) the student nominated all non-deskmates who had nominated them. This minimized the number of friendships between deskmates and maximized the number of friendships between non-deskmates. Second, we re-ran the central analyses with dyadic non-response weights. The resulting estimates identify the causal effect of interest under the assumptions that our non-response model is correctly specified. A more detailed description, the full analysis code and results of these additional analyses can be found on the Open Science Framework.

**Software.** All models were estimated in the R [35] software package *brms* [36, 37] using R Studio [38]. We used the default Bayesian priors in *brms*, which are non-informative, or very weakly informative. All figures were created in *ggplot2* [39].

## IRB approval and consent

This study was reviewed and approved by the IRB offices at the Center for Social Sciences, Budapest (data collection and analysis), and at the University of Wisconsin-Madison (data analysis). Consent was obtained at multiple points. School districts, school principals, and teachers provided written consent to participating in the seating chart randomization. Parents provided written consent for the retrieval of administrative records via teachers, and for their children's participation in the survey.

## Results

### Deskmate hypothesis: Effect of the intervention on friendships

We analyzed the effect of being seated next to each other (for the duration of one semester) on students' reciprocated friendships within the classroom (after the end of the semester) using Bayesian multi-membership multilevel probit models. We report results first as probit coefficients for the effects on students' latent continuous propensity toward friendship (Table 1), and second as average marginal effects (AME) for the effects on students' predicted probability of a manifest reciprocated friendship. Since AMEs are non-linear functions of the probit coefficients, they answer different questions and may lead to qualitatively different conclusions (see Methods).

The results show that sitting next to each other had a large positive effect on students' friendships. The intervention increased the latent continuous friendship propensity of a dyad (Table 1, Main Analysis, $b_{Deskmate} = 0.27$; $CI_{95}$: [0.19, 0.35]), and it increased the predicted

**Table 1. Results of Bayesian multi-membership multilevel probit models for the effects of sitting next to each other on reciprocated friendship.**

|  | Main Analysis | | Modification by Overall Dyadic Similarity | |
| --- | --- | --- | --- | --- |
|  | Estimate | 95% CI | Estimate | 95% CI |
| $b_0$ | -0.96 | [-1.20; -0.72] | -1.49 | [-1.84; -1.14] |
| $b_{Deskmate}$ | 0.27 | [0.19; 0.35] | 0.29 | [0.19; 0.39] |
| $\sigma_{Student}$ | 0.04 | [0.00; 0.10] | 0.52 | [0.46; 0.58] |
| $b_{Similarity}$ |  |  | 0.83 | [0.79; 0.86] |
| $b_{Similarity*Deskmate}$ |  |  | 0.07 | [-0.05; 0.18] |
| $N_{Dyads}$ | 24,962 |  | 24,962 |  |
| $N_{Students}$ | 2,996 |  | 2,996 |  |

probability of a manifest friendship by 7.0 percentage points ($CI_{95}$: [4.6; 9.4]), from 15.3 percent to 22.3 percent. This evidence confirms the deskmate hypothesis: induced proximity fostered friendships.

Our conclusions remained unchanged when including students in the analysis who were assigned to sit alone, and when analyzing directed friendship nominations (regardless of reciprocation; see detailed results on the Open Science Framework). Excluding dyads who did not adhere to treatment resulted in a slightly larger effect estimate of 7.2 percentage points ($CI_{95}$: [4.9; 9.8]). In models in which we allowed the deskmate effect to vary between classrooms, we found some variability of the deskmate effects, but the differences were substantively small ($SD_{Deskmate\ effect}$ = 0.09, $CI_{95}$: [0.00, 0.23]), see S1 Fig. These models also suggested that the number of students in the classroom did not modify the deskmate effect (bottom tertile, 17 students or fewer: $AME$ = 7.5 percentage points, $CI_{95}$: [4.5, 10.5], top tertile, more than 20 students: $AME$ = 7.3 percentage points, $CI_{95}$: [4.4, 10.5]).

Imputing missing outcomes in the most conservative manner results in a lower bound estimate of $b$ = 0.17, $CI_{95}$: [0.09, 0.24]). In this model, sitting next to each other increased the probability of a manifest friendship by 4.0 percentage points ($CI_{95}$: [2.0; 6.1]), from 14.6 percent to 18.7 percent. Lastly, applying non-response weights, we estimated that the deskmate effect was $b$ = 0.24, $CI_{95}$: [0.15, 0.33]). In this model, sitting next to each other increased the probability of a manifest friendship by 5.9 percentage points ($CI_{95}$: [3.5; 8.4]), from 14.8 percent to 20.8 percent.

## Modification-by-similarity hypothesis: Moderating role of overall similarity

We documented observational homophily by inspecting the association between reciprocated friendship and Gower's index for dyadic similarity between students [25], which included students' gender (boy vs. girl), educational achievement (baseline grade-point average [GPA]), and ethnicity (Roma vs. non-Roma). As expected, there was a strong association between dyadic similarity and dyads' tendency to form a reciprocated friendship ($b_{Similarity}$ = 0.83 per $SD$ of the similarity index, $CI_{95}$: [0.80, 0.86]; $AME$ = 9.0 percentage points, $CI_{95}$: [8.5, 9.5], from low [- 1 $SD$] to average similarity; $AME$ = 20.03 percentage points, $CI_{95}$: [19.4, 21.4], from average to high [+ 1 $SD$] similarity; $N_{Students}$ = 2,996, $N_{Dyads}$ = 24,962).

We tested the modification-by-similarity hypothesis by asking whether the causal deskmate effect varied with students' dyadic similarity. Support for the modification-by-similarity hypothesis was scale dependent, as is often the case in non-linear probability models (see Methods). We did not find evidence that similarity modified the deskmate effect with respect to the latent continuous friendship propensity (see Table 1, Overall Similarity), and model fit

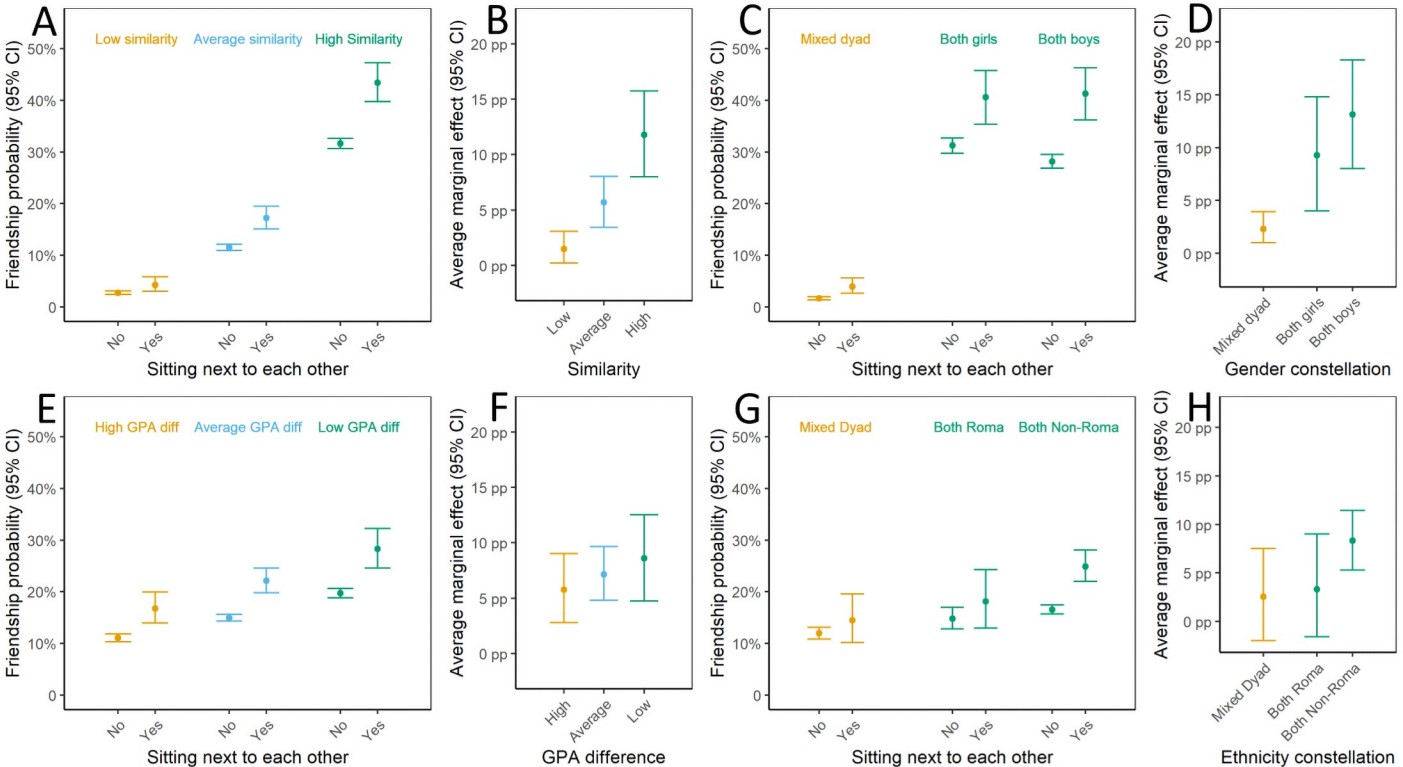

**Fig 1. Model predictions for the effect of sitting next to each other on reciprocated friendships.** Left column displays the predicted friendship probabilities (predictive margins); right column displays the corresponding average marginal deskmate effects (differences in the predicted friendship probabilities) in percentage points. (A) and (B): effect modification by overall similarity (low: -1 SD, high: +1 SD on Gower's similarity index based on gender, ethnicity, and baseline GPA). (C) and (D): effect modification by the gender composition of the dyad. (E) and (F): effect modification by absolute GPA difference between the students (low: -1 SD, high: +1 SD relative to the mean absolute difference). (G) and (H): effect modification by the ethnic (Roma/non-Roma) composition of the dyad.

did not improve when including it ($Diff_{ELPD} = 1.2 = 0.7 {}^{*} SE_{Diff\ ELPD}$ in favor of the more parsimonious model without the interaction term).

By contrast, students' dyadic similarity positively modified the deskmate effect with respect to the probability of manifest friendships (Fig 1A and 1B). The AME of sitting next to each other on the probability of manifest friendships was $AME_{Low}$ = 1.5 percentage points ($CI_{95}$: [0.2, 3.1]) for dyads of low similarity; $AME_{Average}$ = 5.7 percentage points ($CI_{95}$: [3.4, 8.0]) for dyads of average similarity; and $AME_{High}$ = 11.7 percentage points ($CI_{95}$: [7.7, 15.6]) for dyads with high similarity. The 95% credible intervals for the differences between the deskmate effects for dyads with low, average, and high similarity, respectively, comfortably excluded zero ($\Delta AME_{Average-Low}$ = 4.2 percentage points, $CI_{95}$: [0.3, 5.7]; $\Delta AME_{High-Average}$ = 6.1 percentage points, $CI_{95}$: [2.6, 9.8]).

Together, these results indicate that sitting next to each other equally increased the latent continuous friendship propensity for all dyads, and it also increased the probability of manifest friendships, even for fairly dissimilar dyads. But since the baseline friendship propensity was much larger among similar dyads (due to homophily), increasing the friendship propensity by a fixed amount pushed more similar dyads than dissimilar dyads across the threshold of manifest friendship. Thus, seating similar students next to each other resulted in more additional reciprocated friendships than did seating dissimilar students next to each other. Imputing missing values in the most conservative manner did not change conclusions regarding the lack of an interaction on latent friendship propensities. Furthermore, we still observed an

interaction on the probability of manifest friendships (i.e., 95% credible intervals for the differences between the deskmate effects for dyads with low, average, and high similarity exclude zero), but all average marginal effects were somewhat smaller and the 95% credible interval now contained zero for low-similarity dyads: $AME_{Low}$ = 1.7 percentage points ($CI_{95}$: [−0.4, 1.9]); $AME_{Average}$ = 3.1 percentage points ($CI_{95}$: [1.2, 5.1]); and $AME_{High}$ = 7.6 percentage points ($CI_{95}$: [4.0, 11.1]). The same pattern held for analyses applying non-response weights, with average marginal effects falling between the estimates from the pre-registered complete-case analysis and the lower bound analysis: $AME_{Low}$ = 1.1 percentage points ($CI_{95}$: [−0.1, 2.7]); $AME_{Average}$ = 4.8 percentage points ($CI_{95}$: [2.5, 7.2]); and $AME_{High}$ = 10.6 percentage points ($CI_{95}$: [6.5, 15.0]).

## Modification-by-similarity hypothesis: Moderating influence of gender, educational achievement and ethnicity

To better understand the modifying role of dyadic similarity for the effect of proximity on friendship, we performed separate follow-up analyses along each dimension of similarity. For the estimated coefficients, see S1 Table. A more detailed summary of the results, including all estimates, credible intervals, and model comparisons can be found on the Open Science Framework.

**Gender.** Results closely mirrored the results for the overall similarity index. There was strong associational homophily: Same-gender dyads ($N_{both\ boys}$ = 6,700, $N_{both\ girls}$ = 5,848) were much more likely to report a reciprocated friendship than mixed-gender dyads ($N_{mixed\ gender}$ = 12,414). Evidence for the modification-by-similarity hypothesis was again scale dependent, with no modification of the effect of induced proximity on the latent continuous friendship propensity, but clear differences in the effects of sitting next to each other on the probability of manifest friendships. This can be seen in Fig 1C and 1D. Being seated next to each other increased the probability of a friendship among mixed-gender dyads by $AME_{mixed\ gender}$ = 2.3 percentage points ($CI_{95}$: [1.0, 3.9]), among all-female dyads by $AME_{both\ girls}$ = 9.3 percentage points ($CI_{95}$: [4.0, 14.8]); and among all-male dyads by $AME_{both\ boys}$ = 13.1 percentage points ($CI_{95}$: [8.0, 18.3]). Imputing missing values in the most conservative manner, as well as non-response weighting, led to the same pattern of results (albeit with smaller effect estimates).

**Educational achievement.** There was once again clear evidence for associational homophily: the larger the absolute difference in baseline GPA between two students (controlling for dyad's mean baseline GPA), the smaller their propensity to report a reciprocated friendship. With respect to the modification-by-similarity hypothesis, we again found no effect modification with respect to the latent continuous friendship propensity. Furthermore, the intervention increased the probability of a manifest friendship for dyads in which students had similar or dissimilar grades (Fig 1E). While the AMEs of sitting next to each other on the probability of a manifest friendship increased slightly with the similarity of students' grades (Fig 1F), the 95% credible intervals for comparisons between the AMEs computed at different levels of dyadic similarity included zero. Once again, imputing missing values in the most conservative manner, as well as non-response weighting, led to the same pattern of results, with overall smaller effect estimates.

**Ethnicity.** Again, we observed associational homophily: ethnically-matched dyads ($N_{both\ Non-Roma}$ = 16,811, $N_{both\ Roma}$ = 2,851) had a higher latent propensity for reciprocated friendships than dyads of mixed ethnicity ($N_{Mixed\ ethnicity\ dyad}$ = 3,932). And again, we found no evidence that the ethnic constellation of the dyad modified the effect of sitting next to each other on the latent continuous friendship propensity. Considering the effects on manifest friendships, we found ambiguous support for effect modification by ethnic match. There was

some rather weak evidence that the average marginal effect was higher for non-Roma dyads than for mixed ethnicity dyads ($AME_{mixed\ ethnicity\ dyad} - AME_{both\ Non-Roma}$ = 5.8 percentage points, $CI_{95}$: [0.1, 11.2]; see Fig 1G and 1H, although the upper bound of the $CI_{95}$ crossed zero in two alternative model specifications, reported on the Open Science Framework). There was no evidence that the AME for Roma dyads was higher than the AME for mixed-ethnicity dyads ($AME_{both\ Roma} - AME_{mixed\ ethnicity\ dyad}$ = 0.8 percentage points, $CI_{95}$: [-6.4, 8.2]). Taken together, this is at best ambiguous evidence for the modification-by-similarity hypothesis with respect to ethnicity, which would predict that the deskmate intervention is more effective at promoting friendships among *both* non-Roma and Roma dyads compared to dyads of mixed ethnicity. Imputing missing values, as well as non-response weighting, led to the same somewhat unclear pattern of results.

## Discussion

We executed a large pre-registered field experiment that randomized the seating charts in 182 3rd through 8th grade classrooms. We found clear evidence for a positive causal effect of proximity on friendship: sitting next to each other at the beginning of the semester substantially increased the probability of students' mutual best-friendship nominations after the semester had ended. This reverts the Null finding of the only recent, non-randomized, proximity intervention among school-age children [20, 21].

Crucially, our study contributes nuanced new findings regarding the interactions between proximity and homophily in friendship formation. First, replicating prior findings about the importance of homophily as a descriptive characteristic of friendship networks, we established that friendships were more likely to occur between students who shared the same gender, similar levels of academic achievement, and the same ethnicity. Next, we newly investigated the extent to which similarity modified the causal effect of being seated next to each other. Encouragingly, we found no evidence that induced proximity affected the latent continuous propensity towards friendship differentially for similar or dissimilar dyads of students. But since the effect of a given increase in the latent propensity toward friendship on the formation of a manifest friendship also depends on the dyad's baseline propensity toward friendship, and since more similar dyads have a greater baseline propensity toward friendship (homophily), the intervention was more successful among similar students than among dissimilar students. One potential explanation could be that being seated next to each other may be particularly effective at preventing the dissolution of pre-existing ties (as compared to inducing new ties), which are more prevalent among similar dyads; our design, however, does not allow for the identification of different possible mechanistic explanations. The three dimensions of similarity that we investigated contributed to the overall pattern to varying degrees: Gender showed the clearest effect modification (smaller effects among mixed-gender dyads), with a weak but aligned trend for baseline GPA (smaller effects when grade differences were large), and a somewhat misaligned trend for ethnicity (smaller effects in mixed *and* in Roma dyads).

Induced spatial proximity nevertheless succeeded in inducing some diverse friendships. Randomly seating boys and girls next to each other doubled their probability of nominating each other as best friends (from less than 2 to 4 percentage points). The intervention also substantially increased friendships between students with strong and weak baseline GPAs (from 11 to 17 percentage points). Finally, whether or not seating Roma and non-Roma students next to each other increased friendships across ethnic lines remained unclear in our data; the estimate was beset with statistical uncertainty due to relatively small numbers of Roma students in the sample and sensitive to assumptions about missing data.

It remains, of course, an open question whether our findings generalize to other settings and countries. Our study took place in a less prosperous area of rural Hungary, where students' standardized reading and math scores fell below the national average, and fewer parents had graduated from college. Furthermore, study participation depended on teachers' and schools' willingness to participate; and it is possible that the included schools share certain features (e.g., a certain degree of openness) that made students more susceptible to the effects of induced proximity. Despite these potential concerns regarding external validity, which naturally arise in field experiments, we consider our findings in this particular setting promising.

We conclude that even small changes in spatial proximity can substantially affect friendships, not only among the strangers studied in previous research, but also in groups that already know each other well. This documents that some friendship networks remain malleable long after intra-group friendships have presumably been established. Furthermore, proximity also increases the propensity towards friendships, and the probability of manifest friendships, that transgress certain socio-cultural group boundaries—even as the transformation of latent propensities into manifest friendships remains to some extent an uphill battle against pervasive homophily.

Re-seating students is a low-cost and scalable intervention. Friendships across gender, achievement, and ethnical divides formed at a young age likely contribute to the development of social skills and shape attitudes with lasting consequences. This suggests the exciting possibility that targeted, low cost, and scalable interventions may reshape social networks to foster positive life outcomes for students, decrease segregation, and improve inter-group relations.

## Supporting information

**S1 Text. Deviations between the pre-analysis plan and the reported analyses.**
(DOCX)

**S2 Text. Details regarding pre-treatment variables.**
(DOCX)

**S3 Text. Balance checks.**
(DOCX)

**S1 Table. Results of Bayesian multi-membership multilevel probit models investigating the modifying role of single dimensions of similarity.**
(DOCX)

**S1 Fig. Heterogeneity of the deskmate effect across classrooms.** Probit coefficients from random effects model (left panel) as well as the corresponding model-implied friendship probabilities for non-deskmates versus deskmates (right panel). The difference between each predicted probability for deskmates minus the predicted probability for non-deskmates is the classroom-specific AME.
(TIF)

## Acknowledgments

We would like to thank Steffen Nestler, Felix Schönbrodt, Alexander J. Etz, Stefan C. Schmukle, Michael Sobel, Benjamin Rosche, and Jingying He for advice. All errors are ours.

## Author Contributions

**Conceptualization:** Tamás Keller, Felix Elwert.

**Data curation:** Tamás Keller.

**Formal analysis:** Julia M. Rohrer, Felix Elwert.

**Funding acquisition:** Tamás Keller, Felix Elwert.

**Investigation:** Tamás Keller, Felix Elwert.

**Project administration:** Tamás Keller.

**Supervision:** Felix Elwert.

**Writing – original draft:** Julia M. Rohrer.

**Writing – review & editing:** Julia M. Rohrer, Tamás Keller, Felix Elwert.

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
