## [Decision Letter · Decision Letter 0]

13 Jan 2021

PONE-D-20-33057

Proximity Can Induce Diverse Friendships: A Large Randomized Classroom Experiment

PLOS ONE

Dear Dr. Rohrer,

Thank you for submitting your manuscript to PLOS ONE. After careful consideration, we feel that it has merit but does not fully meet PLOS ONE’s publication criteria as it currently stands. Therefore, we invite you to submit a revised version of the manuscript that addresses the points raised during the review process.

I agree with the reviewers that the paper covers an interesting topic and it is overall well executed. However, both reviewers have some concerns about your research design and/or the interpretation of the results. Furthermore, they suggest a number of ways to improve further your analysis.

More specifically, I agree with Reviewer #2 that you should provide, if available, ex-ante information on pre-treatment students’ friendship and pre-treatment seating rules. This piece of information is crucial especially for pupils in higher grades. Furthermore, you specify in the paper that the students were randomly seated in three main subjects, but t is not clear how you chose these subjects and, most importantly, what happens during the other subjects. I also recommend to discuss more in detail the issues of unobservable teacher characteristics and sample selection raised by the same reviewer.

Finally, as suggested by Reviewer #1, it may be interesting to investigate the existence of heterogeneous effects by class size.  Regarding heterogeneous effects already discussed in this version of the paper, the reviewer provides a number of suggestions to clarify your results.

We look forward to receiving your revised manuscript.

Kind regards,

Federica Maria Origo

Academic Editor

PLOS ONE

Journal Requirements:

Reviewers' comments:

Reviewer's Responses to Questions

**Comments to the Author**

1. Is the manuscript technically sound, and do the data support the conclusions?

Reviewer #1: Yes

Reviewer #2: Yes

2. Has the statistical analysis been performed appropriately and rigorously? 

Reviewer #1: Yes

Reviewer #2: No

3. Have the authors made all data underlying the findings in their manuscript fully available?

Reviewer #1: Yes

Reviewer #2: No

4. Is the manuscript presented in an intelligible fashion and written in standard English?

Reviewer #1: Yes

Reviewer #2: No

5. Review Comments to the Author

Reviewer #1: This paper reports results from a field experiment conducted on a sample of Hungarian pupils enrolled in primary school. It shows that exogenous induced physical proximity might increase the likelihood of friendship formation. This effect seems stronger when the physical proximity is induced on students with a higher degree of "similarity".

I find this paper quite interesting and generally well-executed. Overall, I see the potential for publication. I have just a couple of comments, which are reported below.

1) I would suggest moving the section "Modification-by-Similarity Hypothesis: Moderating Influence of Gender, Educational Achievement and Ethnicity" before section "Modification-by-Similarity Hypothesis: Moderating Role of Overall Similarity". I feel it is better to understand first the characteristics driving the effects' heterogeneity and then an overall assessment via a comprehensive measure of similarity.

2) Looking at the results for each dimension of similarity separately, it seems that gender is the only characteristic that matters in altering the primary effect of being a deskmate. Indeed, the grouped GPA and ethnicity marginal effects are never statistically significantly different from each other (Fig 1, H and F). This aspect should also be clarified in the text. For instance, the last sentence in the abstract gives the impression that ethnicity seems to be a relevant character but is not that clear from the results.

3) With the class fixed effects, you control for the unobservables related to the class. However, could you highlight how the effect changes depending on the size of the class-size? Is the effect stronger in larger classes? It should be appropriate to run an interaction model with an indicator for sample size (for instance, being in the top or bottom tertile of class-size distribution), but given how the results are presented, it could also work a sample splitting.

Reviewer #2: Is the manuscript technically sound, and do the data support the conclusions?

I have some concerns on the design:

- ex-ante information: we miss information on students’ ex-ante friendship relationship within the class. This would have allowed a much cleaner design and test of the research hypotheses. Do the authors have such information?

- how are students’ seat decided usually? Do students choose? Are they in alphabetical order? Do the authors have information on the previous seating scheme within each class?

- why do the authors choose only three subjects? I imagine they are the subjects who represent most of the teaching hours. But, wouldn’t be better to ask to fix students’ seat for all the teaching subjects? It is not clear what happens during the other subjects. Do students change their seating when the subject changes?

Has the statistical analysis been performed appropriately and rigorously?

I list here some concerns:

- selection: it can happen at different levels. First, line 127, recruitment depends on teachers’ decision to implement the protocol. Teachers deciding to take part to the study may have unobserved characteristics that also influence pupils attitudes towards - let’s call them - “several kinds” of friendship. Second, some schools are then dropped because they did not meet the inclusion criteria; third students and schools are dropped from the sample because they did not answer the friendship-nomination item which is used to create the outcome variable. The authors should show that selection is not an issue. The first step in this direction would be to compare observable characteristics across samples.

- can the authors check the robustness without students with self reported measures?

- line 149: have you checked the robustness of results without such 5.6%

- line 329: can the authors comment on the size of the effect? 1.6 compared with 7 percentage points seems quite a relevant difference

- the authors can include equations of the estimated models and tables with the estimation results. This would help to understand their econometric technique and make the reading of the paper more fluent

Is the manuscript presented in an intelligible fashion and written in standard English?

- the authors can include equations of the estimated models and tables with the estimation results. This would help to understand their econometric technique and make the reading of the paper more fluent

- Please, spellcheck the paper because I have spotted some typos: i.e. line 154 “students”; line 155 “with”; line 250 “be friend”; line 461 “replicating”

6. PLOS authors have the option to publish the peer review history of their article (what does this mean?). If published, this will include your full peer review and any attached files.

Reviewer #1: No

Reviewer #2: No

---

## [Author Response · Author response to Decision Letter 0]

23 Feb 2021

[Please refer to the Word Document that we uploaded which is properly formatted to ensure readability]

Point-by-point response

Editorial remarks:

E.1. I agree with Reviewer #2 that you should provide, if available, ex-ante information on pre-treatment students’ friendship and pre-treatment seating rules. This piece of information is crucial especially for pupils in higher grades. 

We have revised the manuscript to clarify that teachers typically design the seating chart. We provide additional information on typical seating practices in our response to R.2.2 below. 

We do not have ex-ante information students’ prior friendships or past seating charts. We agree that information on prior friendships (i.e., lagged outcomes), or prior deskmates (i.e., lagged treatments) would have been interesting, as it would have allowed us to answer additional research questions, and may potentially have increased efficiency. 

However, the absence of this information is not a problem for the causal claims we make: We are able to identify the causal effects of having a particular deskmate because we randomized the seating chart. These causal effects are the central focus of our manuscript. Thus, believe that our study makes valuable contributions since it is the first large randomized experiment on the effect of spatial proximity within classrooms on friendships.

E.2. Furthermore, you specify in the paper that the students were randomly seated in three main subjects, but it is not clear how you chose these subjects and, most importantly, what happens during the other subjects.

Thank you for this suggestion! Indeed, this is important background information. We have expanded the relevant manuscript section to include it.

p. 8, starting from line 163: Teachers were instructed to employ the intended seating chart in the three core subjects of the curriculum—mathematics, Hungarian literature, and Hungarian grammar—from the first day of classes (September 1, 2017) until the end of the fall semester (January 31, 2018). These three subjects form the core of the curriculum and receive the greatest weight in admission to selective secondary schools. They were taught in the same room for all grade levels and accounted for 6 to 10 lessons per week (25 to 45 percent of all lessons). Enforcing the seating chart across all subjects was not possible because (1) in some subjects, classrooms were split into smaller groups (e.g., different foreign languages), and (2) in some subjects, students were not seated in a fixed grid-layout (e.g., physical education and arts). However, seating charts typically apply to all subjects in a given room, and (depending on the grade level) most subjects were taught in the same room. Thus, students assigned to sit next to each other in the three core subjects likely also sat next to each other in other subjects; but we did not verify adherence outside of the core subjects.

The decision to limit the intervention instructions to the three core subjects was thus made for the sake of practicality. (We expect that non-adherence to the seating-chart in other subjects would, if it were random, dilute the effects of our intervention and render our estimates conservative. However, since we have no formal result for this expectation, we chose not to include it in the manuscript.) 

E.3. I also recommend to discuss more in detail the issues of unobservable teacher characteristics and sample selection raised by the same reviewer.

We address this question below in response to R.2.4, in order to present our answer in the context of the verbatim reviewer comment.

E.4. Finally, as suggested by Reviewer #1, it may be interesting to investigate the existence of heterogeneous effects by class size. Regarding heterogeneous effects already discussed in this version of the paper, the reviewer provides a number of suggestions to clarify your results.

Done! Please see our detailed response below (R.1.3). 

E.5. If applicable, we recommend that you deposit your laboratory protocols in protocols.io to enhance the reproducibility of your results.

We are commited to research transparency and reproducibility. We were unaware of the protocols.io platform, which seems like a great fit for laboratory research in particular. For the present project, we have already made all relevant information available on the Open Science Framework (https://osf.io/4vjc5/?view_only=d0e9a887b3da4ebcabd0f9afb7480d65). We hope this is acceptable. 

E.6. Please ensure that your manuscript meets PLOS ONE’s style requirements, including those for file naming. 

We have carefully checked all materials to ensure that they adhere to the style requirements.

E.7. Please provide additional details regarding participant consent. In the ethics statement in the Methods and online submission information, please ensure that you have specified what type you obtained (for instance, written or verbal, and if verbal, how it was documented and witnessed). If your study included minors, state whether you obtained consent from parents or guardians. If the need for consent was waived by the ethics committee, please include this information.

Done! We have provided the following details regarding participant consent in the revised manuscript:

p. 16, line 353: This study was reviewed and approved by the IRB offices at the Centre for Social (data collection and analysis), and at the University of Wisconsin-Madison (data analysis). Consent was obtained at multiple points. School districts, school principals, and teachers provided written consent to participating in the seating chart randomization. Parents provided written consent for the retrieval of administrative records via teachers, and for their children’s participation in the survey.

E.8. We note that you have stated that you will provide repository information for your data at acceptance. Should your manuscript be accepted for publication, we will hold it until you provide the relevant accession numbers or DOIs necessary to access your data. If you wish to make changes to your Data Availability statement, please describe these changes in your cover letter and we will update your Data Availability statement to reflect the information you provide.

We are fully committed to provide the data and all other relevant information on the Open Science framework. All materials can already be accessed under https://osf.io/4vjc5/?view_only=d0e9a887b3da4ebcabd0f9afb7480d65. We will make the corresponding project public upon acceptance. 

Reviewer #1

This paper reports results from a field experiment conducted on a sample of Hungarian pupils enrolled in primary school. It shows that exogenous induced physical proximity might increase the likelihood of friendship formation. This effect seems stronger when the physical proximity is induced on students with a higher degree of “similarity”.

I find this paper quite interesting and generally well-executed. Overall, I see the potential for publication. I have just a couple of comments, which are reported below.

Thank you for this positive feedback.

R.1.1. I would suggest moving the section “Modification-by-Similarity Hypothesis: Moderating Influence of Gender, Educational Achievement and Ethnicity” before section “Modification-by-Similarity Hypothesis: Moderating Role of Overall Similarity”. I feel it is better to understand first the characteristics driving the effects’ heterogeneity and then an overall assessment via a comprehensive measure of similarity.

We see the merit of this suggestion—the “bottom-up” logic (start with single dimensions, then aggregate) may be easier to follow for some readers. On the other hand, our pre-analysis plan states that the combined index of similarity is of primary interest; the single dimensions are only listed as secondary (exploratory) analyses. Thus, we have respectfully elected to keep the current structure in order to to honor the pre-registered sequence of primary vs. secondary analyses. We thank the reviewer for this thoughtful idea.

R.1.2. Looking at the results for each dimension of similarity separately, it seems that gender is the only characteristic that matters in altering the primary effect of being a deskmate. Indeed, the grouped GPA and ethnicity marginal effects are never statistically significantly different from each other (Fig 1, H and F). This aspect should also be clarified in the text. For instance, the last sentence in the abstract gives the impression that ethnicity seems to be a relevant character but is not that clear from the results.

We agree with the reviewer’s interpretation that gender is the main driver of modification by similarity (on the average-marginal-effect [AME] scale). As for the other two dimensions, the story is somewhat complicated. For example, the pairwise comparisons of the effect by GPA-category may not be statistically significant in themselves, but they align with the trend higher similarity � larger AME; hence this GPA trend will still contribute to the modification by overall similarity.

As requested, we have sharpened the prose, including in the abstract, to be crystal clear. 

Abstract: […] the probability of a manifest friendship increased more among similar than among dissimilar students—a pattern mainly driven by gender. Our findings demonstrate that a scalable light-touch intervention can affect face-to-face networks and foster diverse friendships in groups that already know each other, but they also highlight that transgressing boundaries, especially those defined by gender, remains an uphill battle.

p. 22, line 491: But since the effect of a given increase in the latent propensity toward friendship on the formation of a manifest friendship also depends on the dyad’s baseline propensity toward friendship, and since more similar dyads have a greater baseline propensity toward friendship (homophily), the intervention was more successful at inducing manifest friendships among similar students than among dissimilar students. The three dimensions of similarity that we investigated contributed to this pattern to varying degrees: Gender showed the clearest effect modification (smaller effects among mixed-gender dyads), with a weak but aligned trend for baseline GPA (smaller effects when grade differences were large), and a somewhat misaligned trend for ethnicity (smaller effects in mixed and in Roma dyads).

R.1.3. With the class fixed effects, you control for the unobservables related to the class. However, could you highlight how the effect changes depending on the size of the class-size? Is the effect stronger in larger classes? It should be appropriate to run an interaction model with an indicator for sample size (for instance, being in the top or bottom tertile of class-size distribution), but given how the results are presented, it could also work a sample splitting.

Thank you for suggesting this interesting additional analysis. We derived the requested estimates from the model in which the effect was allowed to vary freely from classroom to classroom. These exploratory findings are now briefly summarized in the manuscript: 

p. 17, line 381: In models in which we allowed the deskmate effect to vary between classrooms, we found some variability of the deskmate effects, but the differences were substantively small (SDDeskmate effect = 0.09, CI_95: [0.00, 0.23]), see S3 Fig. These models also suggested that the number of students in the classroom did not modify the effect (bottom tertile, 17 students or fewer: AME = 7.5 percentage points, CI_95: [4.5, 10.5], top tertile, more than 20 students: AME = 7.3 percentage points, CI_95: [4.4, 10.5]).

Reviewer #2

R.2.1. I have some concerns on the design:

- ex-ante information: we miss information on students’ ex-ante friendship relationship within the class. This would have allowed a much cleaner design and test of the research hypotheses. Do the authors have such information?

This is a great question. Unfortunately, we do not have information on prior friendships within classes. While such information would have allowed us to ask additional questions and possibly have increased power (maybe this is what the reviewer means by a “cleaner test”), we respectfully submit that this information is not necessary to justify the claims we make in this study. Randomization of the seating chart on its own is sufficient to cleanly identify the causal effect of deskmates on friendship formation. (See also response to E1).

R.2.2. how are students’ seat decided usually? Do students choose? Are they in alphabetical order? Do the authors have information on the previous seating scheme within each class?

From a survey of classroom teachers (N = 160) prior to the intervention, we know that (a) 74% of teachers design the seating chart in their classrooms, (b) some teachers prefer to assign high and low ability students (48.8%), or well and badly behaved students (41.3%), to the same desk. According to teachers’ answers, students’ gender and ethnicity are not important considerations when desiging the seating chart: 75% and 95.6% of teachers reported that these characteristics do not play a role in designing the seating chart. We do not have information on prior deskmate relationship of these students; this information would have allowed us to ask additional questions, but it is not necessary to justify the claims we make in this study—the causal effects we estimate are identified by randomization. 

We have revised the body of the text to include that (p. 4, line 91) “ordinarily, the majority of seating charts would be designed by teachers, see S2 Text.” S2 Text contains the entire paragraph above.

R.2.3. why do the authors choose only three subjects? I imagine they are the subjects who represent most of the teaching hours. But, wouldn’t be better to ask to fix students’ seat for all the teaching subjects? It is not clear what happens during the other subjects. Do students change their seating when the subject changes?

This is an important question—and one that we pondered extensively during the design phase of the study. 

We fully agree that it would have been optimal to fix students’ seating charts for all subjects. But this was not feasible in practice, given that students sometimes change rooms for different subjects. 

We now provide this additional information in the revised manuscript: 

p. 8, starting from line 163: Teachers were instructed to employ the intended seating chart in the three core subjects of the curriculum—mathematics, Hungarian literature, and Hungarian grammar—from the first day of classes (September 1, 2017) until the end of the fall semester (January 31, 2018). These three subjects form the core of the curriculum and receive the greatest weight in admission to selective secondary schools. They were taught in the same room for all grade levels and accounted for 6 to 10 lessons per week (25 to 45 percent of all lessons).. Enforcing the seating chart across all subjects was not possible because (1) in some subjects, classrooms were split into smaller groups (e.g., different foreign languages), and (2) in some subjects, students were not seated in a fixed grid-layout (e.g., physical education and arts). However, seating charts typically apply to all subjects in a given room, and (depending on the grade level) most subjects were taught in the same room. Thus, students assigned to sit next to each other in the three core subjects likely also sat next to each other in other subjects; but we did not verify adherence outside of the core subjects.

R.2.4. selection: it can happen at different levels. First, line 127, recruitment depends on teachers’ decision to implement the protocol. Teachers deciding to take part to the study may have unobserved characteristics that also influence pupils attitudes towards - let’s call them - “several kinds“of friendship. Second, some schools are then dropped because they did not meet the inclusion criteria; third students and schools are dropped from the sample because they did not answer the friendship-nomination item which is used to create the outcome variable. The authors should show that selection is not an issue. The first step in this direction would be to compare observable characteristics across samples.

We believe that this comment addresses multiple types of selection (cf. Imai, King, Stuart. 2008—“Misunderstandings between experimentalists and observationalists about causal inference” JRSS-A). 

Points 1 (teachers) and 2 (schools) concern “selection” on baseline characteristics. Selection on baseline characteristics is regrettable but par for the course in field experiments, which, to an extent, rely on subjects’ willingness to participate in an intervention. This type of selection only concerns external validity (i.e., generalizability or transportability of results across contexts). It does not threaten the internal validity, i.e., identification of causal effects within the study sample. The trade-off between internal and external validity in favor of achieving internal validity is standard in field experiments (Imai et al. 2008), although it goes without saying that it would be preferable to have both (Imai et al. 2008).

Selection on the outcome (friendship nominations), had it occurred, by contrast, would additionally threaten internal validity (i.e., identification). We are optimistic that this problem, should it exist, is minor in our study. First, we emphasize that all exclusion criteria were pre-registered prior to the receipt of outcome data. This prevents us from “fishing” for desired results. Second, outcome data are missing only for a small share of the sample (391 students, 10.3% of the pre-registered 3,814 students), a share that is in line with other well-regarded randomized field experiments. Third, most missingness in the outcome is owed to lack of parental consent to participate in the endline survey, rather than due to item non-response. One would have to craft very elaborate scenarios to link parental lack of consent to bias in the main analysis. Specifically, it would have to be the case that parental consent for the endline survey is a function of the outcome, i.e., whether or not their child had befriended their deskmate (and even in such a scenario, our analysis would be a valid test of the null hypothesis of no effect). That is not to say that we can rule out any threat of bias; only that we do not think this problem is of special concern for our study.

R.2.5. can the authors check the robustness without students with self reported measures?

We assume that this question referrs to the fact that, following the pre-analysis plan, missing teacher reports of baseline covariates were filled in with students’ self-reports collected at endline (line 189). Note that this decision was pre-registered, and it only affected a small number of students (depending on the subject, between 3.2 and 3.6% of the grades were filled in from self-reports). Nonetheless, to ensure that our results weren’t sensitive to this decision, we re-ran analyses limiting the sample to students for which teacher reports were available.

In brief, results were highly similar. The estimated average marginal effect of the intervention was 7.2 percentage points, 95% CI: [4.8; 9.6] as opposed to 7.0 percentage points, 95% CI: [4.6; 9.4]. Considering effect modification by similarity, once again results were virtually unchanged, for example: AME among low similarity students 1.7 [0.3; 3.3] as compared to 1.5 [0.2; 3.1]; among average similarity students 5.9 [3.5; 8.3] as compared to 5.7 [3.4; 8.0]; among high similarity students 11.6 [7.8; 15.6] as compared to 11.8 [8.0; 15.7]. 

We can thus be confident that the decision to rely on student self-reports (where necessary) did not affect conclusions. The full analytic output can be found on the Open Science Framework: https://osf.io/sbn6h/.

R.2.6. line 149: have you checked the robustness of results without such 5.6%

We apologize for not understanding this question. Neither on Line 149, nor anywhere in the manuscript or supplement do we refer to the number “5.6.” In the supplemental text, we state that 5.5% of dyads lacked information on ethnicity. These dyads have already been excluded from analysis (as stated in the supplement).

R.2.7. line 329: can the authors comment on the size of the effect? 1.6 compared with 7 percentage points seems quite a relevant difference

We believe this to be a misunderstanding—1.6 compared to 7 percentage points would indeed be quite relevant. However, as we state in the manuscript, this “1.6” refers to the largest discrepancy between estimates. We have clarified the phrasing, line 346: “The largest absolute difference in the estimated AMEs across models was 1.6 percentage points”

To provide the full context for this largest observed difference:

In the focal models (specification as reported in the manuscript), we estimate that the deskmate effect among boy-dyads is 41.28 – 28.14 = 13.14 percentage points (numbers rounded, more precise output reported on the OSF). For gender-mixed dyads, the deskmate effect is 4.02 - 1.69 = 2.33 percentage points. The estimated difference between these deskmate effects is 10.8 percentage points, 95% Credible Interval: 5.48, 16.15.

In our linear probability models, these estimates look slightly different. The deskmate effect among boy-dyads is 42.06 – 27.88 = 14.18 percentage points; among gender-mixed dyads it is 2.94 – 1.15 = 1.80. The estimated difference between these deskmate effects is 12.4 percentage points, 95% Credible Interval: 7.85, 17.10.

The difference between the differences estimated by the two model specifications—12.4 percentage points versus 10.8 percentage points—is 1.6 percentage points, the largest observed discrepancy. We think that this deviation is rather unsurprising given that (1) we would expect the probit model and the linear probability model to behave differently close to zero and given that (2) differences in differences are estimated with rather large uncertainty (see wide credible intervals). 

Hence, this small discrepancy does not affect our qualitative conclusions. 

R.2.8. the authors can include equations of the estimated models and tables with the estimation results. This would help to understand their econometric technique and make the reading of the paper more fluent

Thank you for this suggestion. We have revised the manuscript to state the main Bayesian multi-membership multilevel model in standard econometric notation, see page 10:

We model the effect of sitting next to each other on reciprocated friendship nominations using a Bayesian multi-membership multilevel probit model. This is a dyad-level model with one observation for each unordered dyad consisting of students i and j in classroom c,

〖Friendship〗_({ij}c)^*=β_0+β_D*〖Deskmate〗_({ij}c)+〖Classroom〗_c+∑_(s∈{i,j})▒〖Student〗_sc +ϵ_({ij}c) (1)

where Friendship_({ij}c)^* is the latent continuous friendship propensity of the dyad; 〖Deskmate〗_({ij}c)=1 if the students in the dyad are deskmates and =0 otherwise; Classroom_c is a vector of classroom fixed effects to account for randomization within classrooms; and ϵ_({ij}c)~N(0,σ_ϵ^2) is the i.i.d. dyad-specific error term. The term ∑_(s∈{i,j})▒〖Student〗_sc refers to the two i.i.d. random effects for the students s in the dyad, Student_sc~N(0,σ_Student^2 ). The latent continuous friendship propensity is linked to manifest friendship via the threshold function Friendship_ijc=1 if Friendship_ijc^*>0 and =0 otherwise.

Furthermore, we added a table with estimation results of the two primary analyses. We also added S5 Table with estimation results of the follow-up analyses of the single dimensions of similarity.

Table 1. Results of Bayesian multi-membership multilevel probit models for the effects of sitting next to each other on reciprocated friendship.

 Main Analysis Modification by Overall Dyadic Similarity

 Estimate 95% CI Estimate 95% CI

b0 -0.96 [-1.20; -0.72] -1.49 [-1.84; -1.14]

bDeskmate 0.27 [0.19; 0.35] 0.29 [0.19; 0.39]

σStudent 0.04 [0.00; 0.10] 0.52 [0.46; 0.58]

bSimilarity 0.83 [0.79; 0.86]

bSimilarity*Deskmate 0.07 [-0.05; 0.18]

NDyads 24,962 24,962

NStudents 2,996 2,996

This table lists probit coefficient. Average marginal effects were derived from the fitted probit model following the procedure described in the Methods section. We decided to retain the figure presenting the AMEs and believe that this dual-presentation strategy (Table and Figure) offers a good balance for readers from a broad variety of fields.

R.2.9. Please, spellcheck the paper because I have spotted some typos: i.e. line 154 “students“; line 155 “with“; line 250 “be friend“; line 461 “replicating”

Thank you for your careful reading of our manuscript! We have fixed the mistakes in (original) line 154, 155, and 461. Our use of the intransitive verb “to befriend” in line 250 is correct. We have given the entire manuscript another careful read to correct remaining mistakes.

---

## [Decision Letter · Decision Letter 1]

6 Apr 2021

PONE-D-20-33057R1

Proximity Can Induce Diverse Friendships: A Large Randomized Classroom Experiment

PLOS ONE

Dear Dr. Rohrer,

Thank you for submitting your manuscript to PLOS ONE. After careful consideration, we feel that it has merit but does not fully meet PLOS ONE’s publication criteria as it currently stands. Therefore, we invite you to submit a revised version of the manuscript that addresses the points raised during the review process.

While one of the two reviewers is happy with how you dealt with her main comments in this version of your paper, the other reviewer has raised a couple of concerns that I fully share.

The main issue that can threat your identification strategy remains self-selection, which may be influenced by treatment status. The reviewer provides an illuminating example on this. If you do not have data to provide further robustness checks on this issue, you should at least discuss whether and how this may influence your estimates. Similarly, you should discuss more clearly the role of pre-existing friendships in driving your results.

We look forward to receiving your revised manuscript.

Kind regards,

Federica Maria Origo

Academic Editor

PLOS ONE

Reviewers' comments:

Reviewer's Responses to Questions

**Comments to the Author**

1. If the authors have adequately addressed your comments raised in a previous round of review and you feel that this manuscript is now acceptable for publication, you may indicate that here to bypass the “Comments to the Author” section, enter your conflict of interest statement in the “Confidential to Editor” section, and submit your "Accept" recommendation.

Reviewer #1: All comments have been addressed

Reviewer #2: (No Response)

2. Is the manuscript technically sound, and do the data support the conclusions?

Reviewer #1: Yes

Reviewer #2: Partly

3. Has the statistical analysis been performed appropriately and rigorously? 

Reviewer #1: Yes

Reviewer #2: No

4. Have the authors made all data underlying the findings in their manuscript fully available?

Reviewer #1: (No Response)

Reviewer #2: Yes

5. Is the manuscript presented in an intelligible fashion and written in standard English?

Reviewer #1: Yes

Reviewer #2: Yes

6. Review Comments to the Author

Reviewer #1: (No Response)

Reviewer #2: I think that the authors have done a good job in addressing the reviewers’ comments and the revised version of the paper is much clearer.

I still have some comments:

- about selection: first, I think that, if the authors do not want or can show the observable characteristics of the teachers and schools that selected out of the study, at least they should discuss the external validity of the study; second, as regards selection on the outcome variable, my concern is that selection may be influenced by the treatment status. Just as an example, it may happen that parents whose child was not happy with the intervention because he could not establish a good bond with the deskmate, are more likely to avoid giving consent. This in turn may be more likely to happen among dissimilar pairs. For this reason, it is important to see the characteristics of these observations and make sure that parents’ decision to deny consent (or generally the missings in the outcome) is not related to treatment status.

- related to the above point and to the identification of the effect: I was asking about pre-existing friendships because it is more likely that, within a class, pupils tend to befriend similar peers. Thus, the strongest effect for similar peers may be due to higher likelihood of a pre-existing bond. Since the authors do not have such information, they should acknowledge this caveat when describing their design and above all their results.

- I better explain the question in R.2.6 (I apologise for the mistake, it was line 169): the authors state “94.4 percent of the dyads in which students actually sat next to each after the second week of classes comprised students who were supposed to sit next to each in the intended seating chart” (line 180). Given that the authors have information on compliance, are the results robust (stronger?) if the authors exclude the 5.6% (100-94.4) of the dyads who were not compliant. What if they exclude also the dyads in which students did not actually sat next to each?

- I suggest again to spellcheck the paper: line 207 “1if”; line 512 “the transformations of latent propensities into manifest friendships remains”

7. PLOS authors have the option to publish the peer review history of their article (what does this mean?). If published, this will include your full peer review and any attached files.

Reviewer #1: No

Reviewer #2: No

---

## [Author Response · Author response to Decision Letter 1]

21 May 2021

(Please refer to the uploaded file for more readable formatting)

Editorial Remarks

E.1. The main issue that can threat your identification strategy remains self-selection, which may be influenced by treatment status. The reviewer provides an illuminating example on this. If you do not have data to provide further robustness checks on this issue, you should at least discuss whether and how this may influence your estimates. 

Reviewer 2 brings up two potential issues of selection: (1) study participation (i.e., selection prior to baseline which does not threaten internal validity but may affect external validity) and (2) selective attrition (i.e., selection after baseline), which may threaten internal validity.

Concerning (1), we added a paragraph to the body of the text that explicitly acknowledges potential threats to external validity, see response to R.2.2. 

Concerning (2), we ran extensive additional analyses to investigate and address the matter. Despite low levels of attrition, we did indeed find some evidence for selective attrition, as suggested by the reviewer. We thus ran two additional sets of analyses: lower bound analyses in which missing responses were imputed under the most conservative assumptions; and analyses incorporating non-response weights. Both resulted in somewhat lower point estimates than the 7.0 percentage-points estimate in our pre-registered primary analyses of complete cases: our lower bound estimate is 4.0 percentage points, and our non-response weighted estimate is 6.1 percentage points. In both cases, the 95% CI still comfortably excludes zero. More details can be found in our response to R.2.3. We have adjusted the language throughout the manuscript at multiple points to acknowledge these new and smaller estimates (see also response to R.2.3). 

E.2. Similarly, you should discuss more clearly the role of pre-existing friendships in driving your results.

Thanks to the helpful clarification by the reviewer, we now understand that the reviewer’s remark does not relate to the identification of the causal deskmate effect, which is the focus of our study, but to the interpretation of the role of similarity, which is an effect modifier (i.e., it indexes effect heterogeneity in the causal deskmate effect). In short, the interpretation implied by the reviewer is fully compatible with our interpretation (and we added it to the manuscript), see response to R.2.4. for details. 

Reviewer #2

R.2.1. I think that the authors have done a good job in addressing the reviewers’ comments and the revised version of the paper is much clearer.

Thank you very much!

R.2.2. About selection: first, I think that, if the authors do not want or can show the observable characteristics of the teachers and schools that selected out of the study, at least they should discuss the external validity of the study

We did not collect extensive data on the teachers and schools who were asked to participate in the study and thus cannot provide a comprehensive analysis of potential limitations of external validity. However, we have school-level data of the schools who decided to participate, which we can compare to all other Hungarian primary schools to give a better idea of the setting of the study. 

Of course, the limited external validity of experimental studies always merits a reminder, and so we added a paragraph to the discussion:

p. 24, line 552: “It remains, of course, an open question whether our findings generalize to other settings and countries. Our study took place in a less prosperous area of rural Hungary, where students’ standardized reading and math scores fell below the national average, and fewer parents had graduated from college. Furthermore, study participation depended on teachers’ and schools’ willingness to participate; and it is possible that the included schools share certain features (e.g., a certain degree of openness) that made students more susceptible to the effects of induced proximity. Despite these potential concerns regarding external validity, which naturally arise in field experiments, we consider our findings in this particular setting promising.”

R.2.3 Second, as regards selection on the outcome variable, my concern is that selection may be influenced by the treatment status. Just as an example, it may happen that parents whose child was not happy with the intervention because he could not establish a good bond with the deskmate, are more likely to avoid giving consent. This in turn may be more likely to happen among dissimilar pairs. For this reason, it is important to see the characteristics of these observations and make sure that parents’ decision to deny consent (or generally the missings in the outcome) is not related to treatment status.

Thank you very much for this clarifying explanation. Such selective dropout seems possible and would indeed threaten the identification of the effect of interest (although it is not quite straightforward to gauge the impact, as the students who did not answer the questionnaire are also part of non-deskmate dyads).

We thus ran additional analyses to (1) assess the selectivity of attrition, and (2) to estimate the deskmate effect taking attrition into account. In short, we did find some evidence for selective attrition. Additional analyses that account for selective attrition in various ways (including a worst-case analysis under extreme assumptions) return somewhat smaller estimates of the deskmate effects, although the corresponding 95% credible intervals still comfortably exclude zero. 

We added the following to the manuscript:

Methods section, p. 15, line 349: “Attrition. About 10% of students were omitted from our main analysis, because they did not provide friendship nominations (e.g., because they lacked parental consent for the endline survey, did not attend school on the day of the assessment, or skipped the question). Multivariate non-response models indicated some selective non-response. While gender and ethnicity did not predict missingness (p > .12), a 1 SD increase in GPA, the model predicted a 2.4 percentage point increase in the probability of response (p = .001), and a 1 SD increase in similarity (Gower’s index) predicted a small but statistically significant decrease of 1.5 percentage points in the probability of response (p = .004). To address possible bias from selective attrition, we ran two additional sets of analyses. 

First, we estimated a lower bound for the deskmate effect by imputing missing friendships nominations under extremely conservative assumptions: whenever nominations were missing, we assumed that (1) the student did not nominate their deskmate and (2) the student nominated all non-deskmates who had nominated them. This minimized the number of friendships between deskmates and maximized the number of friendships between non-deskmates. 

Second, we re-ran the central analyses with dyadic non-response weights. The resulting estimates identify the causal effect of interest under the assumptions that our non-response model is correctly specified. A more detailed description, the full analysis code and results of these additional analyses can be found on the Open Science Framework.”

On the OSF, we additionally provide the following details:

Additional Analyses Attrition: […] Second, we calculated (dyadic) non-response weights. Using a probit model that parallels our central analyses, we predicted whether or not dyads’ friendship status was missing from whether or not they were deskmates, baseline covariates (all three combinations of ethnicity, all three combinations of gender, mean dyad GPA and GPA difference), as well as the interaction between the deskmate indicator and the covariates, and classroom fixed effects. From this model, we predicted response weights, and re-ran the central analyses weighting observations with the inverse of the response probabilities. The resulting estimates identify the causal effect of interest under the assumptions that our non-response model is correctly specified. The full analysis code and results of these additional analyses can be found on the Open Science Framework.

Furthermore, the new results are reported throughout the manuscript in the respective sections:

Results section, Deskmate Hypothesis, p. 18, line 407: “Imputing missing outcomes in the most conservative manner results in a lower bound estimate of b = 0.17, CI_95: [0.09, 0.24]). In this model, sitting next to each other increased the probability of a manifest friendship by 4.0 percentage points (CI_95: [2.0; 6.1]), from 14.6 percent to 18.7 percent. Lastly, applying non-response weights, we estimated that the deskmate effect was b = 0.24, CI_95: [0.15, 0.33]). In this model, sitting next to each other increased the probability of a manifest friendship by 5.9 percentage points (CI_95: [3.5; 8.4]), from 14.8 percent to 20.8 percent.”

Results section, Modification-by-Similarity Hypothesis: Moderating Role of Overall Similarity, p. 20, line 457: Imputing missing values in the most conservative manner did not change conclusions regarding the lack of an interaction on latent friendship propensities. Furthermore, we still observed an interaction on the probability of manifest friendships (i.e., 95% credible intervals for the differences between the deskmate effects for dyads with low, average, and high similarity exclude zero), but all average marginal effects were somewhat lower and the 95% credible interval now contained zero for low-similarity dyads: AME_Low=1.7 percentage points (CI_95:[-0.4,1.9]); AME_Average=3.1 percentage points (CI_95:[1.2,5.1]); and AME_High=7.6 percentage points (CI_95:[4.0,11.1]). The same pattern held for analyses applying non-response weights, with average marginal effects falling between the estimates from the complete cases analysis and from the lower bound analysis: AME_Low=1.1 percentage points (CI_95:[-0.1,2.7]); AME_Average=4.8 percentage points (CI_95:[2.5,7.2]); and AME_High=10.6 percentage points (CI_95:[6.5,15.0]).

Results section, modification by gender: Imputing missing values in the most conservative manner, as well as non-response weighting, led to the same pattern of results (albeit with smaller effect estimates).

Results section, modification by educational achievement: Once again, imputing missing values in the most conservative manner, as well as non-response weighting, led to the same pattern of results, with overall smaller effect estimates.

Results section, modification by ethnicity: Imputing missing values, as well as non-response weighting, led to the same somewhat unclear pattern of results. 

We believe that the results from these additional analyses warrant some qualifications to the way we present our results. We changed parts of the discussion where effect modification is discussed to accommodate the more conservative estimates.

p. 24, line 544: “Induced spatial proximity nevertheless succeeded in inducing some diverse friendships. Randomly seating boys and girls next to each other doubled their probability of nominating each other as best friends (from less than 2 to 4 percentage points). The intervention also substantially increased friendships between students with strong and weak baseline GPAs (from 11 to 17 percentage points). Finally, whether or not seating Roma and non-Roma students next to each other increased friendships across ethnic lines remained unclear in our data; the estimate was beset with statistical uncertainty due to relatively small numbers of Roma students in the sample and sensitive to assumptions about missing data.” 

R.2.4. related to the above point and to the identification of the effect: I was asking about pre-existing friendships because it is more likely that, within a class, pupils tend to befriend similar peers. Thus, the strongest effect for similar peers may be due to higher likelihood of a pre-existing bond. Since the authors do not have such information, they should acknowledge this caveat when describing their design and above all their results.

Thank you for this helpful clarification. We initially thought that the reviewer was concerned about the causal identification of the deskmate effect; we now believe that the reviewer is wondering about the mechanism that may explain effect modification (effect heterogeneity) of the deskmate effect by similarity. We apologize for our earlier misunderstanding.

The reviewer’s hypothesis strikes us as plausible. 

Suppose, for example, that the intervention of being seated next to each other may be highly effective in preventing the dissolution of existing friendships, but less effective in inducing new friendships. If similar students are more likely to have pre-existing friendships, then our finding that sitting next to each other, on net, increases the probability of friendship more among similar dyads than among dissimilar dyads could be explained by the greater probability of pre-existing friendships among similar dyads. Assuming that this reasoning describes the actual data-generating mechanism, then pre-existing friendships would be a mediator of the causal effects of similarity on friendship at endline.

This account does not impinge on the causal interpretation of the deskmate effect (either of the average effect, or of the subgroup effects conditional on baseline similarity). We struggled to understand this particular point because we started from the premise that our design does not allow us to causally identify the effects of similarity (as explained in the Method section). But of course, the term “effect modification” may still be evocative of certain types of causal stories. To avoid any such misinterpretation, we re-read the discussion section, took greater care when communicating the effect modification issue, and also included the possible and plausible interpretation suggested by the reviewer.

p. 23, line 531: “But since the effect of a given increase in the latent propensity toward friendship on the formation of a manifest friendship also depends on the dyad’s baseline propensity toward friendship, and since more similar dyads have a greater baseline propensity toward friendship (homophily), the intervention was more successful among similar students than among dissimilar students. One potential explanation could be that being seated next to each other may be particularly effective at preventing the dissolution of pre-existing ties (as compared to inducing new ties), which are more prevalent among similar dyads; our design, however, does not allow for the identification of different possible mechanistic explanations. The three dimensions of similarity that we investigated contributed to the overall pattern to varying degrees:…”

R.2.5. I better explain the question in R.2.6 (I apologise for the mistake, it was line 169): the authors state “94.4 percent of the dyads in which students actually sat next to each after the second week of classes comprised students who were supposed to sit next to each in the intended seating chart” (line 180). Given that the authors have information on compliance, are the results robust (stronger?) if the authors exclude the 5.6% (100-94.4) of the dyads who were not compliant. What if they exclude also the dyads in which students did not actually sat next to each?

Thank you for this clarification! We re-ran analyses excluding the deskmate dyads who did not adhere to the treatment (i.e., dyads who were assigned to sit next to each other but didn’t do so, and dyads who were not assigned to sit next to each other but did do so). 

The resulting estimates were indeed stronger than the results of the main pre-reported analysis reported in the manuscript. In the analysis that excluded these dyads, students had a 22.6% probability of being friends with a deskmate (vs. 22.3% in our main analysis) and a 15.3% probability of being friends with a non-deskmate (vs. 15.3%), the average marginal effect of being seated next to each other was 7.2 [4.9; 9.8] percentage points (vs. 7.0, [4.6; 9.4]). 

We now briefly mention these additional results in the main body of the text:

p. 18, line 399: “Excluding dyads who did not adhere to treatment resulted in a slightly larger effect estimate of 7.2 percentage points (CI_95: [4.9; 9.8]).”

R.2.6. I suggest again to spellcheck the paper: line 207 “1if”; line 512 “the transformations of latent propensities into manifest friendships remains”

Thank you very much for carefully reading the manuscript and catching these typos. We again spellchecked the whole manuscript and caught two superfluous whitespaces.

---

## [Decision Letter · Decision Letter 2]

12 Jul 2021

Proximity Can Induce Diverse Friendships: A Large Randomized Classroom Experiment

PONE-D-20-33057R2

Dear Dr. Rohrer,

We’re pleased to inform you that your manuscript has been judged scientifically suitable for publication and will be formally accepted for publication once it meets all outstanding technical requirements.

Kind regards,

Federica Maria Origo

Academic Editor

PLOS ONE

Additional Editor Comments (optional):

Reviewers' comments:

Reviewer's Responses to Questions

**Comments to the Author**

1. If the authors have adequately addressed your comments raised in a previous round of review and you feel that this manuscript is now acceptable for publication, you may indicate that here to bypass the “Comments to the Author” section, enter your conflict of interest statement in the “Confidential to Editor” section, and submit your "Accept" recommendation.

Reviewer #1: All comments have been addressed

Reviewer #2: All comments have been addressed

2. Is the manuscript technically sound, and do the data support the conclusions?

Reviewer #1: (No Response)

Reviewer #2: Yes

3. Has the statistical analysis been performed appropriately and rigorously? 

Reviewer #1: (No Response)

Reviewer #2: Yes

4. Have the authors made all data underlying the findings in their manuscript fully available?

Reviewer #1: (No Response)

Reviewer #2: Yes

5. Is the manuscript presented in an intelligible fashion and written in standard English?

Reviewer #1: (No Response)

Reviewer #2: Yes

6. Review Comments to the Author

Reviewer #1: (No Response)

Reviewer #2: (No Response)

7. PLOS authors have the option to publish the peer review history of their article (what does this mean?). If published, this will include your full peer review and any attached files.

Reviewer #1: No

Reviewer #2: No

---

## [Editor Report · Acceptance letter]

16 Jul 2021

PONE-D-20-33057R2 

Proximity Can Induce Diverse Friendships: A Large Randomized Classroom Experiment 

Dear Dr. Rohrer:

I'm pleased to inform you that your manuscript has been deemed suitable for publication in PLOS ONE. Congratulations! Your manuscript is now with our production department. 

Kind regards, 

on behalf of

Dr. Federica Maria Origo 

Academic Editor

PLOS ONE